# A Polyol-Polyol Super-Carbon-Chain Compound Containing Thirty-Six Carbon Stereocenters from the Dinoflagellate *Amphidinium gibbosum*: Absolute Configuration and Multi-Segment Modification

**DOI:** 10.3390/md18120590

**Published:** 2020-11-26

**Authors:** Wan-Shan Li, Zeng Luo, Yan-Lan Zhu, Yi Yu, Jun Wu, Li Shen

**Affiliations:** 1School of Pharmaceutical Sciences, Southern Medical University, 1838 Guangzhou Avenue North, Guangzhou 510515, China; liwanshan@hainnu.edu.cn (W.-S.L.); luozeng1996@i.smu.edu.cn (Z.L.); zyl1120@i.smu.edu.cn (Y.-L.Z.); 2Marine Drugs Research Center, College of Pharmacy, Jinan University, 601 Huangpu Avenue West, Guangzhou 510632, China; yiyu1993@stu2018.jnu.edu.cn; 3Institute of Marine Biomedicine, Shenzhen Polytechnic, Shenzhen 518055, China

**Keywords:** marine dinoflagellate, *Amphidinium gibbosum*, super-carbon-chain compound, absolute configuration, multi-segment modification

## Abstract

A super-carbon-chain compound, named gibbosol C, featuring a polyoxygenated C_70_-linear-carbon-chain backbone encompassing two acyclic polyol chains, was obtained from the South China Sea dinoflagellate *Amphidinium gibbosum*. Its planar structure was elucidated by extensive NMR investigations, whereas its absolute configurations, featuring the presence of 36 carbon stereocenters and 30 hydroxy groups, were successfully established by comparison of NMR data of the ozonolyzed products with those of gibbosol A, combined with *J*-based configuration analysis, Kishi’s universal NMR database, and the modified Mosher’s MTPA ester method. Multi-segment modification was revealed as the smart biosynthetic strategy for the dinoflagellate to create remarkable super-carbon-chain compounds with structural diversity.

## 1. Introduction

Marine dinoflagellates produce huge organic molecules, particularly super-carbon-chain compounds (SCCCs), which are complex natural products with numerous carbon stereocenters on a long-carbon-chain backbone [1,2,3,4]. According to the structural feature of backbones, SCCCs can be categorized into two classes, viz., polyol-polyene and polyol-polyol compounds.

To date, most reported SCCCs are polyol-polyene compounds, such as amphidinol homologs [5,6,7,8,9,10,11,12,13,14,15,16] and karlotoxin congeners [17,18,19,20,21,22], characterized by the presence of both a polyol and a polyene chain, connected by a central core containing two tetrahydropyran rings. Polyol-polyene compounds, mainly isolated from marine dinoflagellates of the genera *Amphidinium* and *Karlodinium*, exhibit antifungal, antitumor, antiosteoclastic, and analgesic effects [5,6,7,8,9,10,11,12,13,14,15,16,17,18,19,20,21,22]. These SCCCs can specifically bind to membrane cholesterol or ergosterol and then disrupt cell membranes without altering their integrity [23].

Polyol-polyol compounds, however, are SCCCs featuring the presence of two polyol chains connected by a central core containing tetrahydropyran rings. So far, few examples have been reported. Ostreol B, isolated from the marine dinoflagellate *Ostreopsis* cf. *ovata*, could be classified as a polyol-polyol compound, although it only contains a tetrahydropyran ring as the central core [24]. Previously, two SCCCs with activation or inhibitory effects on the expression of vascular cell adhesion molecule 1, named gibbosols A and B (Figure 1a), were obtained by our team from the South China Sea dinoflagellate *Amphidinium gibbosum*. After years of unremitting effort, the planar structures and absolute configurations of these SCCCs have been completely established by a combined chemical, spectroscopic, and computational approach [25]. Gibbosols A and B represent a new type of polyol-polyol SCCC. During our ongoing investigation of SCCCs from the same dinoflagellate, a minor polyol-polyol SCCC, named gibbosol C (**1**) (3.0 mg) (Figure 1b), was obtained by the aid of the UPLC-MS/MS technique. The structures of the C-1–C-4, C-11–C-17, and C-34–C-35 segments within the starting polyol chain of gibbosol C (**1**) were different from those of gibbosol A. In this work, the isolation, planar structure elucidation, and stereochemical assignment of gibbosol C (**1**) are reported.

## 2. Results and Discussion

### 2.1. Planar Structure and Ozonolysis of Gibbosol C (***1***)

The molecular formula of **1** was established as C_76_H_142_O_32_ with six degrees of unsaturation by the positive high resolution-electrospray ionization-mass spectrometry [(+)-HR-ESI-MS)] ions at *m/z* 784.4814 ([M + 2H]^2+^, calcd 784.4815) and 1567.9579 ([M + H]^+^, calcd 1567.9557). According to the ^1^H and ^13^C NMR data of **1** (Table 1), four degrees of unsaturation result from four carbon–carbon double bonds. Thus, the molecule must contain two rings. The ^13^C NMR data and the DEPT135 experiment of **1** indicated the presence of 76 carbon resonances that can be categorized as seven methyl groups, 25 methylene groups (including an oxymethylene), 41 methine groups (including 5 olefinic and 33 oxymethine groups), and three non-protonated olefinic carbons.

Three substructures, viz., **I** (from C-1 to C-13, C-71, C-72, C-73, and C-74, in orange), **II** (from C-14 to C-33, C-75, and C-76, in green), and **III** (from C-34 to C-70, in pink), were determined by analysis of key ^1^H–^1^H COSY, H2BC, and HMBC correlations of **1** (Figure 1c).

For substructure **I**, the linear connection of C-1 to C-4 was indicated by the proton spin system H_2_-1–H-2–H_2_-3–H_2_-4, which was deduced from the corresponding ^1^H–^1^H COSY correlations; H2BC correlations from H-2/C-1 and H-2/C-3; and HMBC correlations from H-2/C-4 and H_2_-1/C-3. Similarly, the linear connection of C-6 to C-10 and the branched connection of C-9 and C-72 were revealed by the proton spin system H_2_-6–H-7–H_2_-8–H-9(H_3_-72)–H-10, H2BC correlations from H-7/C-6 and H-7/C-8, and HMBC correlations from H-7/C-9, H_3_-72/C-8, H_3_-72/C-9, and H_3_-72/C-10. The linear connection of C-4 to C-6 and the branched connection of C-5 and C-71 were corroborated by the H2BC correlation from H-6/C-5 and HMBC cross-peaks from H_3_-71/C-4, H_3_-71/C-5, H_3_-71/C-6, and H-7/C-5, whereas the linear connection of C-10 to C-13 and the branched connections from C-11/C-73 and C-13/C-74 were confirmed by HMBC cross-peaks from H-10/C-12, H_3_-73/C-10, H_3_-73/C-11, H_3_-73/C-12, H_3_-74/C-12, and H_3_-74/C-13. Taken together, the planar structure of **I** was elucidated (Figure 1c).

For substructure **II**, the linear connection of C-14 to C-16 was revealed by ^1^H–^1^H COSY correlations between H_2_-14/H_2_-15 and H_2_-15/H-16, H2BC correlations between H_2_-14/C-15, and HMBC cross-peaks between H_2_-14/C-16. Similarly, the linear connection of C-16 to C-27 was corroborated by two proton spin systems, H_2_-18–H-19–H_2_-20 and H_2_-22–H-23–H_2_-24–H-25– H-26–H-27; H2BC correlations between H-17/C-16, H-17/C-18, H-19/C-20, H_2_-20/C-21, H_2_-22/C-21, H-23/C-22, H-25/C-24, and H-27/C-26; and key HMBC cross-peaks between H-17/C-15, H-17/C-19, H-19/C-21, H-23/C-21, H-25/C-23, and H-27/C-25. In addition, the linear connection of C-27 to C-33 and the branched connections between C-31/C-75 and C-33/C-76 were established by ^1^H–^1^H COSY correlations between H_2_-30/H-31 and H_3_-75/H-31, H2BC correlations between H-27/C-28, H_2_-30/C-29, and H_2_-30/C-31, and HMBC cross-peaks between H-27/C-29, H-31/C-29, H_3_-75/C-30, H_3_-75/C-31, H_3_-75/C-32, H_3_-76/C-32, and H_3_-76/C-33. Taken together, the planar structure of **II** was assigned (Figure 1c).

For substructure **III**, the linear connection of C-34 to C-70 was undoubtedly elucidated by ^1^H–^1^H COSY, H2BC, and HMBC correlations (Figure 1c). The linear connection of C-34 to C-36 was revealed by ^1^H–^1^H COSY correlations from H_2_-34/H_2_-35 and H_2_-35/H-36, H2BC correlations from H_2_-34/C-35 and H-36/C-35, and HMBC cross-peaks from H_2_-34/C-36. The linear connections of C-39 to C-50, C-52 to C-54, C-56 to C-58, and C-61 to C-70 were confirmed by four proton spin systems, viz., H-39–H-40–H-41–H-42–H_2_-43–H-44–H-45–H-46–H_2_-47–H-48–H_2_-49–H_2_-50, H-52–H-53–H-54, H-56–H-57–H_2_-58, and H-61–H-62–H-63–H_2_-64–H-65–H_2_-66–H-67–H-68–H-69–H_3_-70; H2BC correlations from H-41/C-40, H-42/C-41, H-42/C-43, H-48/C-49, H-56/C-57, H-62/C-61, H-62/C-63, H-65/C-64, H-67/C-66, and H-67/C-68; and HMBC cross-peaks from H-39/C-41, H-42/C-40, H-42/C-44, H-44/C-46, H-46/C-48, H-48/C-50, H-54/C-52, H-56/C-58, H-62/C-64, H-65/C-63, H-67/C-65, H-67/C-69, H_3_-70/C-68, and H_3_-70/C-69. The linear connections of C-37 to C39, C-50 to C52, C-54 to C56, and C-58 to C-61 were resolved by H2BC correlations from H-39/C-38, H_2_-50/C-51, H-52/C-51, H-54/C-55, H-56/C-55, H_2_-58/C-59, and H-61/C-60 and crucial HMBC correlations from H-39/C-37, H-52/C-50, H-53/C-51, H-53/C-55, H-54/C-56, H-57/C-59, H_2_-58/C-60, H-61/C-59, and H-62/C-60 (Figure 1c)

The presence of two tetrahydropyran moieties in **III** was indicated by isotope shift experiments, measured in both CD_3_OD and CD_3_OH [26]. Four oxymethine carbons involved in ether linkage, viz., C-44 (*δ* 70.9), C-48 (*δ* 73.2), C-52(*δ* 76.5), and C-56 (*δ* 75.76), did not exhibit a deuterium-induced isotope shift. The presence of ether bridges between C-44/C-48 and C-52/C-56, respectively, was further supported by HMBC cross-peaks from H-44/C-48 and H-52/C-56. Taken together, the backbone of **III** was assembled (Figure 1c).

Finally, crucial HMBC cross-peaks from H_3_-74/C-14 and H_2_-14/C-12 confirmed the connection between **I** and **II** through C-13−C-14, whereas the correlations from H_3_-76/C-34, H_2_-34/C-32, and H_2_-34/C-33 revealed the connection between **II** and **III** through C-33−C-34. The large value of ^3^*J*_H-68,H-69_ (14.7 Hz) and NOE correlations from H_3_-72/H_3_-73, H-10/H-12, H_3_-75/H_3_-76, H-32/H_2_-34, and H_3_-70/H-68 concluded that the geometries of the four carbon–carbon double bonds in **1**, viz., C10═C11, C12═C13, C32═C33, and C68═C69, were all *E*-configured. Based on the above results, the planar structure of **1**, containing 30 hydroxy groups and six pendant methyl moieties, was successfully established (Figure 1c).

Due to the heavy overlap of the ^1^H and ^13^C NMR signals of **1**, the relative configurations of its stereogenic carbons could not be determined based on various 1D and 2D NMR data of the intact **1**. Comparison of the planar structure of **1** with that of gibbosol A (Figure 1) revealed that the structures of the C-5–C-11, C-17–C-33, and C-37–C-70 segments in **1** are the same as those of the C-4–C-10, C-18–C-34, and C-36–C-69 segments in gibbosol A, respectively. Based on the same biosynthetic machinery, the absolute configurations of the corresponding segments above should be identical. With detailed NMR data of three ozonolyzed products of gibbosol A (viz., gA**a**–**c**, Figure 2a) at hand [25], ozonolysis reaction was carried out for **1** to obtain the corresponding NMR data for comparison. As a result, O_3_/NaBH_4_-mediated cleavage of the carbon–carbon double bonds of **1** afforded three main fragments, viz., **1a**, **1b**, and **1c** (Figure 2b, Appendix A). Both **1b** and **1c** were obtained as epimeric pairs at C-13 and C-33, respectively.

### 2.2. Relative and Absolute Configurations of Gibbosol C (***1***)

Because gibbosols A and C were produced by the same marine dinoflagellate, the common biosynthetic origins of the two SCCCs should lead to identical absolute configurations of the corresponding segments in the three main pairs of the ozonolyzed fragments above. Detailed analysis of the NMR data led to the conclusion that the relative configurations of **1****a**, **1****b**, and **1****c** were similar to those of gA**a**, gA**b**, and gA**c**, respectively, except for the insertion of an additional methylene group between C-2 and C-4 in **1a**, the presence of an additional 16-OH group in **1b** and an additional 36-OH group in **1c**, and the absence of the 13,15-diol and 16-Me (Me-73 in gA**b**) groups in **1b** (Figure 2). Coincidentally, all these modifications appear on the starting segments within three ozonolyzed products of gibbosol A [25].

For the C-5−C-7 segment of **1a**, *J*-based configuration analysis (*J*BCA) [27,28] was used (Figure 3a). ^3^*J*_H,H_ values were measured by 2D *J*-resolved spectroscopy (2D *J*RES), whereas ^2,3^*J*_C,H_ values were obtained by the HETLOC experiment. The anti orientations between H-5/H-6a and H-7/H-6b were assigned by the large values of ^3^*J*_H-5,H-6a_ (10.1 Hz) and ^3^*J*_H-7,H-6b_ (9.5 Hz), respectively, whereas the gauche orientations between H-5/H-6b and H-7/H-6a were proved by the small values of ^3^*J*_H-5,H-6b_ (3.9 Hz) and ^3^*J*_H-7,H-6a_ (3.3 Hz), respectively. The anti orientations between C-4/H-6b and 7-OH/H-6a were deduced from the large value of ^3^*J*_C-4,H-6b_ (6.8 Hz) and the small value of ^2^*J*_C-7,H-6a_ (−1.9 Hz), respectively. The gauche orientations between C-71/H-6a, C-71/H-6b, C-4/H-6a, C-8/H-6a, C-8/H-6b, and 7-OH/H-6b were established by the small values of ^3^*J*_C-71,H-6a_ (3.0 Hz), ^3^*J*_C-71,H-6b_ (2.8 Hz), ^3^*J*_C-4,H-6a_ (1.3 Hz), ^3^*J*_C-8,H-6a_ (2.4 Hz), and ^3^*J*_C-8,H-6b_ (2.8 Hz) and the large value of ^2^*J*_C-7,H-6b_ (−6.6 Hz), respectively. Thus, the relative configuration between Me-71/7-OH in **1a** was concluded to be *syn* (Figure 3a).

The intermediate values of ^3^*J*_H-7,H-8a_ (7.6 Hz) and ^3^*J*_H-7,H-8b_ (5.2 Hz) indicated two interconverting conformations for both H-7/H-8a and H-7/H-8b. Similarly, those of ^2^*J*_C-7,H-8b_ (−3.1 Hz) and ^3^*J*_C-6,H-8a_ (3.6 Hz) suggested two alternating conformations for both 7-OH/H-8b and C-6/H-8a. The large value of ^2^*J*_C-7,H-8a_ (−6.1 Hz) and the small value of ^3^*J*_C-6,H-8b_ (1.1 Hz) revealed that both 7-OH/H-8a and C-6/H-8b remained in gauche orientations. Thus, two alternating conformers were assigned for the C-7−C-8 segment (Appendix A). Similarly, the intermediate values of ^3^*J*_H-9,H-8a_ (6.2 Hz) and ^3^*J*_H-9,H-8b_ (7.2 Hz) indicated two interconverting conformations for both H-9/H-8a and H-9/H-8b. Those of ^3^*J*_C-72,H-8b_ (4.2 Hz) and ^3^*J*_C-10,H-8a_ (3.3 Hz) suggested two alternating conformations for both C-72/H-8b and C-10/H-8a. The small values of ^3^*J*_C-72,H-8a_ (1.8 Hz) and ^3^*J*_C-10,H-8b_ (2.3 Hz) revealed that both C-72/H-8a and C-10/H-8b were in gauche orientations. Therefore, two alternating conformers were assigned for the C-8−C-9 segment. Based on the results above, the relative configuration between 7-OH/Me-72 was determined as *syn* (Figure 3a).

To determine the absolute configurations of C-2, C-7, and C-9 in **1****a**, the modified Mosher’s MTPA method was used. Through a comparison of the sign of Δ*δ^SR^* values between **1as**/**1ar**, the absolute configurations of C-2 and C-7 in **1a** were determined to be *S* and *R* (Figure 3b), respectively [29]. In addition, the absolute configuration of C-9 was assigned as *R* by the widely separated H_2_-10 signals of **1ar** (*δ* 4.29, 4.18) when compared with those of **1as** (*δ* 4.18, 4.13) [30,31]. Based on this *syn* relationship between Me-71 and 7-OH, the absolute configuration of C-5 in **1a** was established as *R*. Therefore, the absolute configurations of the stereogenic carbons in **1a** were established as 2*S*,5*R*,7*R*,9*R* (Figure 3b), which are the same as those in gA**a**.

Based on Kishi’s universal NMR database, the relative configurations of the C-17−C-25 segment in **1b** were assigned as (anti/anti/anti/anti), the same as those of the C-18−C-26 segment in gA**b** (Figure 2) [25], by chemical shifts of three central carbons of the consecutive 1,3,5-triol moieties, viz., C-19 (*δ* 66.5), C-21 (*δ* 66.4), and C-23 (*δ* 66.4) [32,33].

To establish the relative configuration between 16-OH and 17-OH, *J*BCA [27,28] was used. ^2,3^*J*_C,H_ values were obtained by both the HETLOC and HECADE experiments (Figure 4a). Though the observed ^3^*J*_H-16,H-17_, ^2^*J*_C-16,H-17_, and ^2^*J*_C-17,H-16_ values fell into the intermediate range, implying interconversion between two conformers, the gauche orientations between C-18/H-16 and C-15/H-17 were undoubtedly determined by the small values of ^3^*J*_C-18,H-16_ (1.6 Hz from HETLOC and 1.7 Hz from HECADE) and ^3^*J*_C-15,H-17_ (2.3 Hz from HETLOC and 2.5 Hz from HECADE). Accordingly, the relationship between 16-OH and 17-OH was concluded to be *syn* (Figure 4a).

For the C-36−C-37 segment, the gauche orientation between H-36 and H-37 was supported by the small value of ^3^*J*_H-36,H-37_ (2.7 Hz). The gauche orientations between C-38 and H-36 and between C-35 and H-37 were revealed by the small values of ^3^*J*_C-38,H-36_ (1.1 Hz) and ^3^*J*_C-35,H-37_ (2.3 Hz), while the anti orientations between 36-OH and H-37 and between 37-OH and H-36 were deduced from the small values of ^2^*J*_C-36,H-37_ (1.2 Hz) and ^2^*J*_C-37,H-36_ (0.4 Hz). Therefore, the relationship between 36-OH and 37-OH was concluded to be *syn* (Figure 4b).

The relative configurations of the C-37–C-39 segment in **1c** were determined to be the same as that of the C-36–C-38 segment of gA**c** by *J*BCA [27,28] (Figure 4b). For the C-37−C-38 segment, the gauche orientation between H-37 and H-38a was determined by the small value of ^3^*J*_H-37,H-38a_ (4.0 Hz), while the anti orientation between H-37 and H-38b was assigned by the large value of ^3^*J*_H-37,H-38b_ (9.3 Hz). Similarly, the anti orientation between 37-OH and H-38a was established by the small value of ^2^*J*_C-37,H-38a_ (−1.9 Hz), whereas the gauche orientation between 37-OH and H-38b was established by the large value of ^2^*J*_C-37,H-38b_ (−5.5 Hz). In addition, the gauche orientations between C-36 and H-38a and between C-36 and H-38b were suggested by the small values of ^3^*J*_C-36,H-38a_ (1.1 Hz) and ^3^*J*_C-36,H-38b_ (2.3 Hz), respectively. For the C-38−C-39 segment, the intermediate values of ^3^*J*_H-39,H-38a_ (5.6 Hz) and ^3^*J*_H-39,H-38b_ (7.2 Hz) indicated two interconverting conformations for both H-39/H-38a and H-39/H-38b. Similarly, those of ^2^*J*_C-39,H-38a_ (−3.7 Hz) and ^3^*J*_C-40,H-38b_ (3.4 Hz) suggested two alternating configurations for both 39-OH/H-38a and C-40/H-38b. The gauche orientations between 39-OH/H-38b and C-40/H-38a were deduced from the large value of ^2^*J*_C-39,H-38b_ (−5.3 Hz) and the small value of ^3^*J*_C-40,H-38a_ (1.0 Hz). Accordingly, the relationship between 37-OH and 39-OH was concluded to be *syn*. Finally, the relative configurations of the C-36−C-39 segment in **1c** were concluded to be *syn*/*syn* (Figure 4b).

Furthermore, the relative configurations of the C-39−C-42 segment in **1c** were assigned as *syn*/anti/anti (Appendix A), the same as those of the C-38−C-41 segment in gA**c** [25], on the basis of Kishi’s universal NMR database [27,28].

Based on the absolute configurations of the C-17–C-25 and C-37–C-42 segments of **1**, viz., 17*R*,19*S*,21*S*,23*R*,25*S* and 37*S*,39*R*,40*S*,41*R*,42*R*, which were deduced from those of the corresponding segments of gibbosol A, the absolute configurations of C-16 in **1b** and of C-37 in **1c** were assigned as *R* and *S*, respectively. In summary, the absolute configurations of 36 stereogenic carbons in gibbosol C (**1**) were established as 2*S*, 5*R*, 7*R*, 9*R*, 16*R*, 17*R*, 19*S*, 21*S*, 23*R*, 25*S*, 26*R*, 27*R*, 29*R*, 31*S*, 36*S*, 37*S*, 39*R*, 40*S*, 41*R*, 42*R*, 44*R*, 45*S*, 46*R*, 48*R*, 52*R*, 53*S*, 54*S*, 55*R*, 56*R*, 57*R*, 59*S*, 61*R*, 62*R*, 63*R*, 65*R*, and 67*R* (Figure 1b).

In the pathogenesis of atherosclerosis, vascular cell adhesion molecule 1 (VCAM-1) and intercellular adhesion molecule 1 (ICAM-1) promote the accumulation of macrophages within the intima, leading to formation of the atherosclerotic lesion [34,35,36]. The above two adhesion molecules, particularly VCAM-1, have been considered as potential therapeutic targets for anti-atherogenic drug development [37]. It is important to find promising VCAM-1 inhibitors from natural products. Thus, the effects of gibbosol C (**1**) on VCAM-1 and ICAM-1 in human umbilical vein endothelial cells were investigated according to previous procedures [25]. At the concentration range of 10.0 and 100.0 μg/mL, gibbosol C (**1**) showed no obvious activities on VCAM-1 and ICAM-1 expression (Appendix A), whereas gibbosol A displayed remarkable activation effects on VCAM-1 expression. On the contrary, gibbosol B exhibited marked inhibitory activities on VCAM-1 expression [25].

The co-isolation of gibbosols A–C enabled us to summarize their structural features, which may shed light on the biosynthesis of these polyol-polyol SCCCs. The 13 carbon central cores (C-43–C-55); the C-4–C-10, C-18–C-34, and C-36–C-42 segments of the starting polyol chain; and the whole terminal polyol chain (C-56–C-69) of gibbosol A are quite conserved, whereas other segments of the starting polyol chain are variable. Diverse modification patterns, such as insertion, substitution, oxidation, and reduction, appear on the C-1–C-4, C-11–C-17, or C-34–C-35 segment of the starting polyol chain of gibbosol A. In other words, multi-segment modification on the starting polyol chain is the biosynthetic strategy for the generation of gibbosol C (**1**).

In addition, glycolate should be the starter unit in the biosynthesis of gibbosols A–C [38]. Though acetate labeling patterns of some polyol-polyene SCCCs, such as amphidinols A [38], **4 [2,39]**, and **17 [39,40]**, have been reported, no enzymatic mechanisms involved in the biosynthesis of these SCCCs have been uncovered so far [38,39,40,41]. Of course, the acetate labeling patterns of gibbosols A–C are worthy of further investigation in future.

## 3. Materials and Methods

### 3.1. General Experimental Procedures

HR-ESI-MS was obtained on a Bruker maXis ESI-QTOF mass spectrometer (Bruker Daltonics, Bremen, Germany) in the positive-ion mode. LR-ESI-MS was recorded on a Bruker amaZon SL mass spectrometer (Bruker Daltonics, Bremen, Germany) in both the positive- and negative-ion modes. One- and two-dimensional NMR spectra were measured on a Bruker AV-700 MHz NMR spectrometer (Bruker Scientific Technology Co. Ltd., Karlsruhe, Germany). UV spectra were recorded on a UV-2600 UV-Vis spectrophotometer (SHIMADZU, Kyoto, Japan) and optical rotations determined on an MCP 500 modular circular polarimeter (Anton Paar GmbH, Seelze, Germany) with a 1.0 cm cell at 25 °C. Preparative HPLC was performed on a Waters 2535 pump equipped with a YMC C_18_ reversed-phase column (250 × 10 mm i.d., 5 μm, Kyoto, Japan) and a 2998 photodiode array detector coupled with a 2424 evaporative light scattering detector (Waters Corporation, Milford, NY, USA). For column chromatography, silica gel (100–200 mesh, Qingdao Mar. Chem. Ind. Co. Ltd., Qingdao, China) and C_18_ reversed-phase silica gel (ODS-A-HG 12 nm, 50 µm, YMC, Kyoto, Japan) were employed.

### 3.2. Isolation of the Dinoflagellate and the Large-Scale Culture

The isolation and culture of the marine dinoflagellate *Amphidinium gibbosum* was described in our previous publication [25].

### 3.3. Isolation of Gibbosol C (***1***)

The filtrate of the culture medium (1200 L) was loaded onto a macroporous resin column (DIAION, HP-20, 120 cm × 15 cm i.d.), eluted with freshwater to remove sea salt. The loaded sample was successively eluted with 25%, 50%, 75%, and 95% aqueous ethanol. All the eluates were concentrated under reduced pressure to afford the resultant solid (5.5 g), which was separated by a C_18_ reversed-phase column (60 × 5 cm i.d.), eluted with an aqueous methanol solution (10% to 100%) to yield 83 fractions. UPLC-MS (Waters ACQUITY UPLC BEH C_18_ 150 × 2.1 mm i.d., 1.7 μm, MeCN/H_2_O, from 5:95 to 98:2) was used for the detection of super-carbon-chain compounds in these fractions. Fractions 62 and 63 (23.6 mg) were combined and purified by HPLC (YMC-Pack 250 × 4.6 mm i. d., MeCN/H_2_O, 21:79) to give **1** (3.2 mg, *t*_R_ = 60.5 min).

Gibbosol C (**1**): Colorless solid, ${\left[a\right]}_{D}^{25}$ = + 9.3 (*c* = 0.08, methanol); UV (MeCN) λ_max_ (log *ε*) 203.6 (3.9), 230.2 (3.6) nm; for ^1^H and ^13^C NMR spectroscopic data, see Table 1; HR-ESI-MS *m*/*z* [M + 2H]^2+^ (calcd for C_76_H_144_O_32_, 784.4814, found 784.4815) and [M + H]^+^ (calcd for C_76_H_143_O_32_, 1567.9579, found 1567.9557).

### 3.4. Ozonolysis

Gibbosol C (**1**) (3.0 mg, 0.002 mM) was dissolved in a mixture of CH_2_Cl_2_–MeOH (1:1, each 2 mL). Ozone was bubbled into the above solution at –78 °C for 2 min. An excess amount of NaBH_4_ was then added and stirred at –78 °C for 3 h. The reaction mixture was purified by a C_18_ reversed-phase silica gel column (Daisogel, SP-120-50-ODS-B, 3 g, 5.0 cm × 1.0 cm i.d.), eluted with 10 mL of water followed by 15 mL of MeOH, to afford three fractions. The second fraction was purified by HPLC (Cosmosil, HILIC, 250 × 4.6 mm i. d., MeCN/H_2_O, 90:10−50:50) to afford three products, viz., fragments **1a** (0.4 mg, *t*_R_ = 4.9 min), **1b** (0.9 mg, *t*_R_ = 50.8 min), and **1c** (1.3 mg, *t*_R_ = 72.7 min) (Figure 2).

**1a**: For ^1^H and ^13^C NMR spectroscopic data, see Appendix A; LR-ESI-MS *m*/*z* 257.6 [M + Na]^+^ and 491.6 [2M + Na]^+^.

**1b**: For ^1^H and ^13^C NMR spectroscopic data, see Appendix A; LR-ESI-MS *m*/*z* 487.6 [M + H]^+^ and 509.5 [M + Na]^+^.

**1c**: For ^1^H and ^13^C NMR spectroscopic data, see Appendix A; LR-ESI-MS *m*/*z* 869.4 [M + H]^+^ and 891.3 [M + Na]^+^.

### 3.5. Mosher’s MTPA Esters ***1as*** and ***1ar***

The fragment **1a** (0.3 mg) was treated with (*R*)-MTPACl (8.0 μL) in dried pyridine (0.8 mL) at room temperature for 10 h. The reaction mixture was concentrated and purified by HPLC (YMC-Pack 250 cm × 4.6 mm i.d., MeCN/H_2_O, 92:8) to afford the (*S*)-MTPA ester **1as** (1.1 mg). The (*R*)-MTPA ester **1ar** (1.0 mg) was prepared in the same way. With the aid of key ^1^H–^1^H COSY correlations, ^1^H NMR spectroscopic data of **1as** and **1ar** were assigned (Appendix A).

**1as**: For ^1^H NMR spectroscopic data, see Appendix A; LR-ESI-MS *m*/*z* 1116.4 [M + NH_4_]^+^ and 1121.2 [M + Na]^+^.

**1ar**: For ^1^H NMR spectroscopic data, see Appendix A; LR-ESI-MS *m*/*z* 1116.4 [M + NH_4_]^+^ and 1121.3 [M + Na]^+^.

## 4. Conclusions

In summary, a new polyol-polyol SCCC, named gibbosol C, was isolated from the South China Sea dinoflagellate *A. gibbosum*. Its planar structure and absolute configurations, featuring the presence of 36 carbon stereocenters and 30 hydroxy groups, were successfully established by extensive NMR investigations, ozonolysis of the carbon–carbon double bonds, *J*-based configuration analysis, Kishi’s universal NMR database, the modified Mosher’s MTPA ester method, and comparison of the NMR data of the ozonolyzed products with those of gibbosol A. Multi-segment modification seems to be the smart biosynthetic strategy for the dinoflagellate to create remarkable SCCCs with diverse structures. Marine dinoflagellates of the genus *Amphidinium* harbor novel and complex SCCC biosynthetic routes. New integrated chemical, spectroscopic, and computational approaches or intelligent databases should be developed to cope with the stereochemical complexity of SCCCs. Evidently, specific carbon–carbon bond cleavages are an important means for the determination of the relative and absolute configurations of polyol-polyol SCCCs in the future.

## Figures and Tables

**Figure 1 marinedrugs-18-00590-f001:**
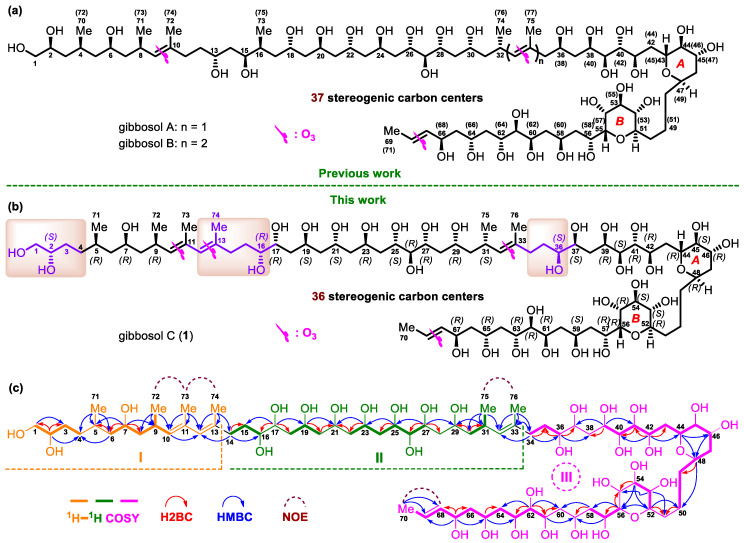
(**a**) The structures and absolute configurations of gibbosols A and B. (**b**) The structure and absolute configurations of gibbosol C (**1**). (**c**) Key ^1^H–^1^H COSY, H2BC, and HMBC correlations and diagnostic NOE interactions of gibbosol C (**1**).

**Figure 2 marinedrugs-18-00590-f002:**
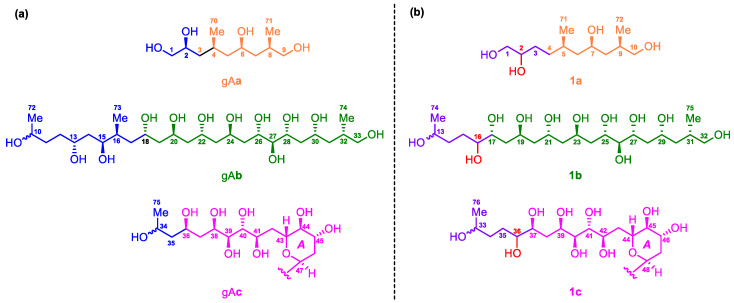
(**a**) Structures of the ozonolyzed fragments gA**a**–**c** of gibbosol A. (**b**) Structures of the ozonolyzed fragments **1a**–**1c** of gibbosol C (**1**).

**Figure 3 marinedrugs-18-00590-f003:**
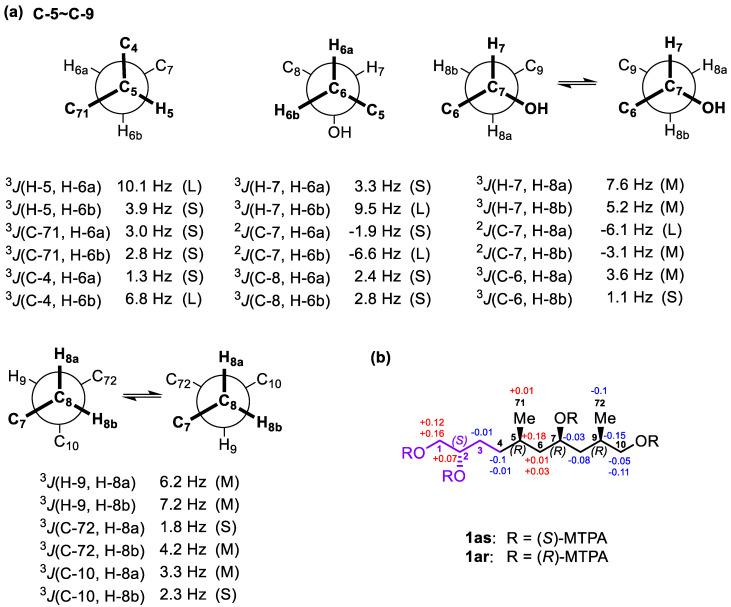
(**a**) Rotamers and coupling constants for the C-5–C-9 segment of **1a**. (**b**) Δ*δ^SR^* values obtained for **1a**.

**Figure 4 marinedrugs-18-00590-f004:**
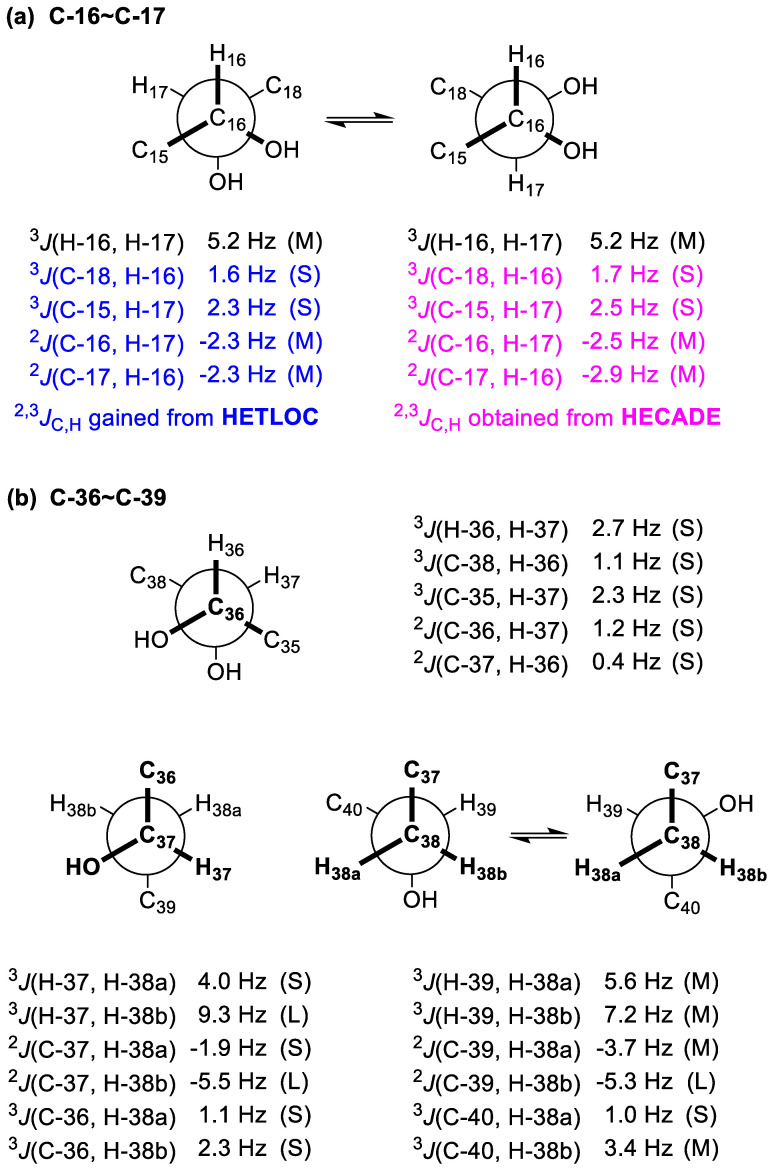
(**a**) Rotamers and coupling constants for the C-16–C-17 segment of **1b**. (**b**) Rotamers and coupling constants for the C36–C39 segment of **1c**.

**Table 1 marinedrugs-18-00590-t001:** ^1^H (700 MHz) and ^13^C (175 MHz) NMR data for **1** in CD_3_OD.

No.	*δ*_H_ (*J* in Hz)	*δ*_C_, Type
1a	3.41, m	67.4, CH_2_
1b	3.45, m	
2	3.53, m	73.6, CH
3a	1.39, m	31.9, CH_2_
3b	1.48, m	
4a	1.32, m	34.8, CH_2_
4b	1.32, m	
5	1.65, m ^f^	30.5, CH
6a	1.12, m	47.1, CH_2_
6b	1.46, m	
7	3.62, m	68.5, CH
8a	1.28, m	47.4, CH_2_
8b	1.43, m ^a^	
9	2.74, m	30.6, CH
10	4.92, br d (9.8) ^i^	136.2, CH
11		133.3, qC
12	5.67, br s	130.3, CH
13		136.3, qC
14a	2.08, m	37.7, CH_2_
14b	2.26, m	
15a	1.47, m	32.3, CH_2_
15b	1.74, m	
16	3.42, m	75.81, CH
17	3.70, m	72.8, CH
18a	1.50, m	41.2, CH_2_
18b	1.67, m ^e^	
19	4.10, m	66.5, CH
20a	1.58, m ^b^	46.9, CH_2_
20b	1.58, m ^b^	
21	4.09, m ^j^	66.4, CH
22a	1.58, m ^b^	46.9, CH_2_
22b	1.58, m ^b^	
23	4.12, m	66.3, CH
24a	1.54, m	41.5, CH_2_
24b	1.79, m	
25	3.88, m	70.5, CH
26	3.37 t, (6.3)	78.8, CH
27	3.83, m	73.1, CH
28a	1.58, m ^b^	41.0, CH_2_
28b	1.80, m	
29	3.78, m	70.2, CH
30a	1.35, m	46.8, CH_2_
30b	1.43, m ^a^	
31	2.69, m	30.1, CH
32	4.92, br d (9.8) ^i^	132.1, CH
33		135.5, CH
34a	2.04, m	37.1, CH_2_
34b	2.19, m	
35a	1.55, m ^c^	32.44, CH_2_
35b	1.69, m	
36	3.44, m	74.7, CH
37	3.69, m	73.5, CH
38a	1.77, m ^g^	37.4, CH_2_
38b	1.81, m	
39	4.13, m	70.36, CH
40	3.46, m	74.6, CH
41	3.69, m	75.7, CH
42	4.04, m	70.06, CH
43a	1.77, m ^g^	34.9, CH_2_
43b	1.98, m	
44	3.67, m	70.9, CH
45	3.03, t (8.4)	77.6, CH
46	3.75, m	70.4, CH
47a	1.70, m	37.6, CH_2_
47b	1.88, m	
48	3.93, m	73.2, CH
49a	1.38, m	32.4, CH_2_
49b	1.91, m	
50a	1.53, m	23.4, CH_2_
50b	1.53, m	
51a	1.52, m ^d^	32.8, CH_2_
51b	1.75, m	
52	3.41, m	76.5, CH
53	3.14, t (7.7)	75.2, CH
54	3.72, t (7.7)	75.1, CH
55	3.78, m	73.8, CH
56	3.60, dd (9.1, 4.9)	75.76, CH
57	4.39, m	67.7, CH
58a	1.52, m ^d^	42.1, CH_2_
58b	1.85, m ^h^	
59	4.09, m ^j^	67.1, CH
60a	1.67, m ^e^	42.7, CH_2_
60b	1.85, m ^h^	
61	4.09, m	69.7, CH
62	3.24, dd (7.7, 1.4)	77.1, CH
63	3.82, m	71.9, CH
64a	1.59, m	42.2, CH_2_
64b	1.94, m	
65	4.08, m ^j^	68.7, CH
66a	1.55, m ^c^	45.7, CH_2_
66b	1.65, m ^f^	
67	4.26, m	70.1, CH
68	5.51, ddq (14.7, 6.3, 1.4)	135.9, CH
69	5.66, m	126.4, CH
70	1.68, br d (6.3)	17.9, CH_3_
71	0.88, d (6.3)	19.7, CH_3_
72	0.96, d (6.3)	22.2, CH_3_
73	1.72, br s	17.6, CH_3_
74	1.76, br s	18.1, CH_3_
75	0.94, d (7.0)	22.4, CH_3_
76	1.68, br s	16.7, CH_3_

^a–j^ Overlapped signals assigned by ^1^H–^1^H COSY, HSQC, H2BC, and HMBC spectra without designating multiplicity.

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
