# Peer review of "A Polyol-Polyol Super-Carbon-Chain Compound Containing Thirty-Six Carbon Stereocenters from the Dinoflagellate Amphidinium gibbosum: Absolute Configuration and Multi-Segment Modification"

_marinedrugs, 2020, doi:10.3390/md18120590_

Round 1

Reviewer 1 Report

I have read the submission of Wan-Shan Li et al. describing the isolation and structure elucidation of a new gibbosol metabolite and I hearby tender my findings.

Overall, this is an interesting and well constructed manuscript. The quality of the writing is high and makes the manuscript easy and enjoyable to read. The quality of the data shown in the supplementary information file is also equally high; the authors are to be commended for their excellent work. Well done.

I only have a few minor points that I would like to raise:

  1. Line 20 (abstract), please replace the semi-colon with a comma.
  2. Line 23 (abstract), please change to “… and the modified Mosher’s MTPA…”
  3. Line 32: change to “…backbones, SCCCs can be catergorized…”
  4. Line 38: change exhibited to exhibit.
  5. Line 40: change to “…altering their integrity.”
  6. Line 52: I am not sure how UPLC-MS/MS can be used to “obtain” a natural product. Do the authors mean they used UPLC-MS/MS to guide the isolation of gibbosol C? If so, maybe change to “…was obtained using UPLC-MS/MS to guide the isolation.”
  7. Line 53: change to “..strategy to gibbosol C (1) are reported.”
  8. Please also add some comment regarding how gibbosol C differs from gibbosol A and B within the text to give the reader some context.
  9. Line 65: change to “…including one oxymethylene),…”
  10. Table 1: I note that H-10 and H-32 are both listed with the same entry, and there may be other duplicates. I appreciate that even at 700 MHz, overlap of 1H and 13C resonances is still possible. Can you authors please indicate instances on the table where signals are interchangeable? Also, some comment should be added to the text to indicate instances when overlap could be an issue and how it has been dealt with?
  11. Table 1: Signal 70, the CH3 appears to have the 3 superscripted.
  12. Line 120: The NOE between H3-75 and H3-76 is not of real value for determining the relative configuration between these two centres without further information. Given CH3-76 is attached to an alkene, an NOE between CH3-75 with either the S or R configurations would be expected. Please strengthen this argument.
  13. Line 129 and 140: While I agree that the biosynthetic machinery that makes the gibbosol metabolites will be almost the same, without proof the authors cannot claim that the absolute configurations MUST be the same. I would recommend replacing MUST with SHOULD in both these lines.
  14. Line 142: As written, the statement “…analysis of NMR data led to the conclusion that the absolute configuration of 1a, 1b and 1c were similar…”. How can you use NMR (other than in a chiral environment like either a chiral solvent or with chiral derivatization like Mosher’s method) to determine ABSOLUTE configuration. Do the authors mean relative here?
  15. Line 226: There is an extra space before 46R
  16. Line 234: change to “The thirteen carbon central core…”
  17. I would request more information regarding the bioactivity, and how it differs for gibbosol C relative to gibbosol A and B, be added to give more context.

Author Response

With regard to the comment 8: We added the sentence at Line 53, i.e., The structures of the C-1‒C-4, C-11‒C-17, and C-34‒C-35 segments within the starting polyol chain of gibbosol C (1) were different from those of gibbosol A.

With regard to the comment 10: We revised Table 1 according to the suggestions.

With regard to the comment 12: We added NOE interactions between H-32/H2-34.

With regard to the comment 17: We added a paragraph at Line 233 of the manuscript, along with the Figure S1 in Supporting Information.

In addition, we checked and corrected other words and mistakes.

Reviewer 2 Report

This is a well written paper on the isolation and structure elucidation of a polyol-polyol compound named gibbosol C, isolated from a marine dinoflagellate. The authors has been able to, in a trustworthy and impressive way, to reveal all 36 stereocenters with given absolute configuration. The work rely on their already published work on the two similar compounds named gibbosol A and B. The report is merely on the structure elucidation relying to a large extent on their previous publication (ref. 25). The story of this seems appropriate and with all the data gathered in the supplementary, there is not much to add. A few minor points could still be done to improve the paper:

1) There is a conjugated diene moiety in the structure and a UV should be given. This is not a major issue, but if possible please add this as a matter of principle (reporting all types of data),

2) Although both gibbosol A and B are mentioned and to some extent spectroscopy data are compared, for the reader of this publication it would be very informative to pinpoint the differences of these three compounds. I could suggest that a figure of both gibbosol A and B together with gobbosol C should be included and/or the differences could be pointed out in the text

3) The authors have included just a small section of 4 lines on any biological testing. This was not specifically promising, but based on their previous publication, the authors should in one or two sentences justify why this test was performed and nothing else was done in this direction. 

All together I look forward to see this nice paper published.

Author Response

With regard to the comment 1: We added a high performance liquid chromatogram and the UV spectrum of 1 in Supporting Information.

With regard to the comment 2: We added the structures of gibbosol A and B into Figure 1 of the manuscript.

With regard to the comment 3: We added a paragraph at Line 233 of the manuscript, along with the Figure S1 in Supporting Information.

Reviewer 3 Report

The authors describe and structure elucidate a SCCC compound, gibbosol C.

The paper is well written,  the methods are well described and the results are well reported.

Some minor comments:

line 228-231. The authors investigated for the effect on endothelial cells. From my point of view, it would be nice to have a  comment concerning this behavior e.g. on VCAM-1 expression of gibbosol C, compared to gibbosol A and B (the work from the authors reported on ref 25).

S137-S138: An indication (e.g  A and B) for 13C and dept135 that are displayed would be nice.

Author Response

With regard to bioactivities of gibbosols A-C, we added a paragraph at Line 233 of the manuscript, along with the Figure S1 in Supporting Information.

We also give an indication of all the carbons in 13C NMR and DEPT135 spectra of the fragment 1c at the pages of S137-S138 in Supporting Information.